# Selected Nanomaterials’ Application Enhanced with the Use of Stem Cells in Acceleration of Alveolar Bone Regeneration during Augmentation Process

**DOI:** 10.3390/nano10061216

**Published:** 2020-06-22

**Authors:** Wojciech Zakrzewski, Maciej Dobrzynski, Zbigniew Rybak, Maria Szymonowicz, Rafal J. Wiglusz

**Affiliations:** 1Department of Experimental Surgery and Biomaterial Research, Wroclaw Medical University, Bujwida 44, 50-345 Wroclaw, Poland; wojciech.zakrzewski@student.umed.wroc.pl (W.Z.); zbigniew.rybak@umed.wroc.pl (Z.R.); maria.szymonowicz@umed.wroc.pl (M.S.); 2Department of Conservative Dentistry and Pedodontics, Wroclaw Medical University, Krakowska 26, 50-425 Wroclaw, Poland; maciej.dobrzynski@umed.wroc.pl; 3Institute of Low Temperature and Structure Research, Polish Academy of Sciences, Okolna 2, 50-422 Wroclaw, Poland

**Keywords:** stem cells, nanomaterials, bone augmentation, nanohydroxyapatite

## Abstract

Regenerative properties are different in every human tissue. Nowadays, with the increasing popularity of dental implants, bone regenerative procedures called augmentations are sometimes crucial in order to perform a successful dental procedure. Tissue engineering allows for controlled growth of alveolar and periodontal tissues, with use of scaffolds, cells, and signalling molecules. By modulating the patient’s tissues, it can positively influence poor integration and healing, resulting in repeated implant surgeries. Application of nanomaterials and stem cells in tissue regeneration is a newly developing field, with great potential for maxillofacial bony defects. Nanostructured scaffolds provide a closer structural support with natural bone, while stem cells allow bony tissue regeneration in places when a certain volume of bone is crucial to perform a successful implantation. Several types of selected nanomaterials and stem cells were discussed in this study. Their use has a high impact on the efficacy of the current and future procedures, which are still challenging for medicine. There are many factors that can influence the regenerative process, while its general complexity makes the whole process even harder to control. The aim of this study was to evaluate the effectiveness and advantage of both stem cells and nanomaterials in order to better understand their function in regeneration of bone tissue in oral cavity.

## 1. Introduction

Nowadays, the progress that has been made in dental surgery allows for far developed tissue regeneration in oral cavity and is expected to expand in the nearest future. Because implant dentistry has become a desirable option for replacement of missing teeth, the effectiveness of this technique is mostly dependent on the proper quality and quantity of alveolar bone [1]. The excessive bone loss forbids the placement of dental implants in the ideal prosthetic position [2]. Among many undesirable conditions, bone may be compromised owing to tumour, trauma, periodontal disease, and so on. It was confirmed by Neophytos D. et al. [3] that alveolar bone with a width of 5 mm requires augmentation procedure before successful implant placement.

Bone is biologically privileged tissue, because it has the capacity to undergo regeneration as a part of repair process [4].

There are several bone manipulation techniques that are used to achieve a predictable long-term success for dental implants, and as is later on explained in this study, autologous bone graft still remains the “gold standard” for the process of bone augmentation, as it is characterised by the most effective osteogenic, osteoconductive, osteoinductive, and immunogenic properties [5].

Lack of dental tissue in the alveolar ridge is eventually destructive for either maxilla or mandible, which will be discussed later in this study. When it comes to optimal implant and periodontal aesthetics, preservation of the labial appearance of the alveolar process in frontal region of maxilla and mandible is crucial [6].

It is important to underline that, when there is an insufficient amount of bone to sustain primary and/or secondary implant stability, the alveolar ridge needs to be augmented before placement of the implant. Inadequate alveolar bone height and width often require bone manipulation before, at the time of, or even after the implant surgery. In the case the bone is reduced, but there is enough of it for primary stability of implant, it is possible to directly cover the parts of implant that are still exposed after implant placement with bone graft [7]. In the case of implants, primary stability is crucial, and inability to acquire such a status is one of the most important contraindications for patient implantation.

Among grafts, it is possible to distinguish among the following: autologous grafts, allografts, and xenografts, which will be thoroughly explained in the study.

Alveolar process is a bony ridge that is present on both maxillary and mandibular bone.

The aim of this study is to summarise current research on bone tissue engineering for the clinician with a focus on stem cells, and to review the success of bone augmentation with their help. This work also attempts to confirm the utility of nanomaterials and stem cell-based therapies in acceleration of bone augmentation processes.

## 2. Changes of the Alveolar Process Following Extraction

### 2.1. Degradation Period

Generally, in order to avoid the degradation processes of bone after extraction, and preserve a proper extraction socket architecture, it is recommended to apply the technique of immediate implant placement at the time of extraction. It is commonly known that the bone modelling process occurring in alveolar process after tooth extraction should be avoided. The bone develops together with teeth, which influence its volume and shape. The alveolar bone supporting the teeth is characterised by distinctive features like rapid and continuous remodeling in response to stimuli by force [8,9,10]. According to Bodic F. et al. [11], the alveolar bone is subjected to mechanical loads for only 15–20 min per day. Because both maxilla and mandible are tooth-dependent tissues, after loss of tooth, it reacts with a reduction of alveolar ridge in both apicocoronal and buccolingual dimensions [12,13] owing to compromised blood supply and resorption of thin bundles of bone during healing [14]. The process is called atrophy. Changes in the alveolar ridge quickly result in alterations of soft tissue (gingiva), which is attached directly to the former structure. Major changes in the extraction site tend to occur within the first 12 months after the extraction [12,15], while the loss of height of the alveolar bone occurs during the first 3 months. These processes occur because of the fact that tissue must adapt its mass and structure to changing mechanical demands. In the absence of stimuli, such as forces derived from swallowing and mastication, the alveolar bone undergoes resorption [16]. After a tooth extraction, there is a cascade of inflammatory reactions that are activated, while the extraction socket is temporarily closed by the blood clot. Although tissue integrity is quickly restored, the residual ridge is being formed, which has life-long catabolic remodeling effects on either maxilla or mandible. There is no major difference in bony tissue degradation development when it comes to different regions of extraction sockets in oral cavity.

### 2.2. Healing Period

The alveolar bone healing process is complex, and its effectiveness depends on the efficacy of hosts’ inflammatory response [16]. The bone microarchitecture analysis allows to evolve healing with trabecular thickness, and progressively increase the number over time [10]. The healing process starts within 12 months after tooth extraction [12], while reorganisation of lamina dura takes place throughout its duration. Because oral tissues are located in an environment rich in microorganisms, the healing process is always impaired by the lack of sterility [17]. The alveolar bone healing process usually occurs without histological cartilage formation, while long bone healing is a process of endochondral ossification [18].

Fracture healing is a process in which the restored bone shows a lack of scar tissue and formation of blood clot is a crucial step in order to begin such a healing process [19]. Cascade of reactions following blood clot formation allows effective tissue healing in alveolar bone, because platelets in the clot carry specific growth factors (GFs). Both osteoblasts and osteoclasts have direct contact with lymphocytes, which suggests a regulatory role of the immune system, especially in later stages of the healing process [19]. Non-alveolar and alveolar bones differ in nature of the cells surrounding them. For example, the latter lacks muscle stem cells which play essential role in fracture healing [20,21].

In the later stages of the healing process, the blood clot is being replaced with granulation tissue [22], which then leads to the development of newly formed vessels. When modelling processes commence, at first, apical and lateral walls of the alveolus are restored. Later, the healing process goes toward the centre and the coronal region of the alveolus. According to Scala A et al. [23], it takes around one month to close the extraction socket with a newly formed bone. The process eventually ends with corticalization of the socket and formation of bone marrow.

Residual ridge is an alveolar process that is formed after healing of soft tissues and bone, followed by extractions. Although it is a life-long process, the reduction of bone is most aggressive during the first 6 months. Residual ridge resorption (RRR) depends mostly on the site of the ridge and occurs differently among individuals. The basic structural change in RRR is about reduction of the size of the ridge under mucoperiosteum.

When a structure, for instance, maxillary bone, undergoes stress, it becomes deformed. The mechanical aspect of bone remodelling is mostly associated with Wolffs’s law [24] of bone transformation, which simply says that bone remodels in response to the forces applied, although this explanation describes very briefly such a complex physiological process like bone remodelling.

The biggest amount of bone loss occurs in the horizontal dimension and happens mainly on the facial aspect of the ridge. On the other hand, vertical ridge height occurs most intensely on the buccal aspect [14]. Long-term lack of tooth in the bone results in increased narrowing and shortening of the ridge, which generally relocates palatally/lingually. Stages of alveolar ridge reduction after tooth loss are presented in Figure 1.

## 3. Factors Influencing Alveolar Bone Loss and Regeneration

### 3.1. Bone Loss Factors

After tooth extraction, the alveolar ridge undergoes uneven atrophy processes [22,25]. Although loss of teeth results in naturally irreversible alveolar bone resorption [26], the destructive process of the bone may start even before extraction of the tooth. It may be complicated with gingivitis present, which leads to periodontopathy, endodontic lesions, or trauma injury. After such a situation, further loss of bony tissue owing to extraction may result in severe complications that occur more quickly than in a case in which the bone stays intact before tooth removal.

Bone degradation can be also caused by several metabolic bone diseases, like vitamin D-resistant rickets (VDRR), focal infections, hyperparathyroidism, age-related parietal bone atrophy, or Paget’s disease. VDRR for instance, is a disease affecting mainly dentin, while enamel remains unchanged [27]. Spontaneous pulpal abscesses, which are formed without carious lesions, are detectable. Big tubular clefts in the region of pulpal horns are visible, with submicroscopic defects in the enamel layer, leading to facilitated invasion of bacterial toxins [28]. Increased bacterial invasion of teeth results in accelerated teeth loss, eventually causing accelerated bone resorption.

### 3.2. Bone Regeneration Factors

Despite bone having a mineral nature, it is a vital and dynamic organ. The histogenesis of bone is directly from mesenchymal connective tissue in the intramembranous bone formation process, and from pre-existing cartilage in endochondral bone formation. Following tooth removal, the normal healing process takes approximately 40 days, starting with clot formation and culminating in a socket filled with bone covered by connective tissue and epithelium [29,30]. The biological principles of bone regeneration comprise the following: osteoinduction, osteogenesis, and osteoconduction. Optimization of these processes has been the goal of new materials used in hard tissue engineering. Osteoinduction process allows migration, followed by proliferation of unspecialised connective tissue cells into bone-forming cell lineage [31]. It induces osteogenesis [32] and GFs determine its action [33].

During osteogenesis, a formation of new bone from both Haversian systems and osteoblastic cells of the grafted bone takes place [34,35]. A direct transfer of vita cells to the area that will regenerate new bone occurs.

Osteoconduction, which is the last of the aforementioned principles, focuses on bone growth on a surface. It implies recruitment of non-adult cells and their stimulation to preosteoblasts [36]. The osteoconduction process means bone growth on a surface. It is a phenomenon often seen in the case of bone implants. Because its function is to provide space and substrates for the biochemical and cellular event progressing the bone formation process, it eventually results in osteogenesis [32,37]. After successful implant insertion, proper biologic width and aesthetics should allow for remodelling of the soft tissue and bone, occurring between 6 months and 1 year [38].

### 3.3. Osteoinductive Factors

Bone repair is a multistep process that involves migration, differentiation, and activation of considerable amount of cell types [39,40]. Taking into consideration that bone tissue is highly vascularised, it requires both bone tissues and blood vessels to be formed in a tight integrity [41]. Current bone regenerative strategies pursue mimicking natural bone regeneration. Bone morphogenetic proteins (BMPs) and vascular endothelial growth factors (VEGFs) are two key regulators of osteogenesis and angiogenesis, acting by promoting osteogenic and endothelial differentiation of stem cells, respectively [42,43]. Both factors act synergistically during bone regeneration.

BMPs are one of the most researched and crucial morphogenetic signals coordinating tissue architecture in the whole organism. Having the appropriate concentration and being placed on specific scaffold, they are capable of inducing new bone formation by turning mesenchymal stem cells into chondroblasts and osteoblasts [44]. BMPs are being increasingly used in surgeries.

They belong to the transforming growth factor (TGF) beta superfamily [45]. There are currently at least 20 members of the aforementioned family and, among them, BMP-2 is one of the most common factors in use.

Among the most useful functions of BMPs, one can distinguish among the following: induction of cell replication, chemotaxis, induction of differentiation, anchorage-dependent cell attachment, osteocalcin synthesis/mineralisation [46], and alkaline phosphatase activity [47]. Recent studies confirmed that using recombinant BMP in order to correct bony defects, furcations, and fenestration leads to periodontal regeneration with ankylosis [48]. On the contrary, when BMP-7 was used in the augmentation process, it resulted in serious periodontal increase without ankylosis. BMPs can also be used to alleviate implant wound healing. Rutherford et al. [49] have shown that application of Osteogenic protein-1 (OP-1) around the extraction socket escalated bone growth measured histologically at 3 weeks.

VEGF is the signal protein produced by cells, having the ability of vasculogenesis and angiogenesis. It additionally mediates osteogenesis [50]. Street et al. [40] prove, that localised VEGF delivery is beneficial even for osteoblasts migration and bone turnover. This means that, delivered to a bone defect, it is an effective strategy to accelerate bone healing [51]. Additionally, VEGF has been successfully used to improve maturation of newly formed bone. VEGF belongs to a sub-family of GFs, that is, the platelet-derived growth factor family of cystine-knot GFs. The serum concentration of VEGF increases in chronic hypoxic conditions like diabetes mellitus [52], because it is a part of the system responsible for restoring oxygen supply in the case of inadequate oxygenation of tissues.

## 4. Augmentation Techniques

The augmentation procedure in dentistry is aimed to increase the volume of alveolar bone, particularly when placement of intrabony implant would otherwise be considered problematic [53]. In order to regenerate a sufficient amount of bone to allow successful implant placement, a ridge augmentation technique is recommended. The intramembranous bone formation pathway is used when intraoral bone augmentation techniques are applied by the dental surgeon. The bone augmentation technique, which is used in order to reconstruct different alveolar ridge defects, depends on the horizontal and vertical extent of the defect. Predictability of corrective procedures is influenced by the span of the edentulous ridge and amount of attachment on neighbouring teeth [54].

### 4.1. Guided Bone Regeneration (GBR)

GBR is similar to guided tissue regeneration, but focuses on development of hard tissues. It is a surgical procedure based on using barrier membranes, with or without bone graft/bone substitutes. Bony regeneration by GBR depends on the migration of osteogenic and pluripotential cells to the defect site in bone and exclusion of cells impeding bone formation [55,56]. It is important to underline that, in order to accomplish successful regeneration of a bone defect, the rate of osteogenesis must exceed the rate of fibrogenesis from the surrounding soft tissue [30,57]. The GBR technique requires four principles in order to successfully fill osseous defect-space maintenance for bone in-growth, stability of the fibrin clot to make the uneventful healing possible, exclusion of epithelium and connective tissue to allow space to be filled with bony tissue, and primary wound closure to promote undisturbed healing [58].

The mechanism of GBR is focused on selective in-growth of bone-forming cells into a bone defect region, which is enhanced when adjacent tissue is kept away with a membrane [59]. This additionally allows to protect the wound from both salivary contamination and mechanical disruption.

There are several techniques used in GBR regarding tri-dimensional bony tissue reconstruction. They are all based on packing bone substitutes into the bony defect and covering it with resorbable or non-resorbable membranes.

### 4.2. Bone Augmentation Methods and Precise Implant Placement

The goal of successful augmentation following tooth loss is to allow performing effective prosthetic replacement that is in harmony with the rest of the adjacent natural dentition.

Resorption of alveolar bone is a natural consequence of tooth loss. This process causes clinical problems, especially in terms of aesthetics. In order to overcome the changes in the oral cavity, it is required to carry out treatment that makes it possible to preserve the natural tissue shape, in order to prepare for prosthetic appliance like an implant [60]. The clinical outcome of implant treatment is challenged especially in compromised bones of elderly patients [61].If more alveolar ridge is preserved, it will guarantee optimal implant placement and proper functioning of prosthetic appliance. Nevertheless, nowadays, clinicians are usually faced with the necessity to place implants in the alveolar bone of smaller volume. Such a situation requires the clinician to carry out a proper pre-treatment with augmentation techniques that will promote a more predictable regenerative outcome [54]. As Figure 2 and Figure 3 show, the properties of alveolar bone can be assisted by several methods.

The ideal alveolar ridge width and height make the placement of a natural appearing pontic possible, which provides maintenance of a plaque-free environment [62]. The structural loss of the residual alveolar ridge can occur as a result of tooth extraction, surgical procedures, periodontal disease, or congenital defects [63,64]. In a situation with a bone missing, the overlying soft tissue tends to collapse into the bone defect, making it difficult to recreate oral cavity aesthetics after application of functional prostheses. Alveolar deformities classification is based on quantity of volumetric horizontal and vertical tissue loss within the alveolar process. Such a classification was established to standardise communication between clinicians in the selection and sequencing of reconstructive procedures [65]. It is crucial to thoroughly evaluate the contour of the partially edentulous ridge before starting the process of fixed partial denture fabrication. As presented in Figure 3, according to Seibert, Class I represents bucco-lingual loss of tissue with normal height of ridge. Class II defect is represented by the loss of alveolar height in apico-coronal axis. On the other hand, Class III has a combination of bucco-lingual and apico-coronal loss of tissue. According to this classification, the bone augmentation technique is dependent on the horizontal and vertical extent of the defect.

Class I: bucco-lingual loss of tissue with normal ridge height in an apicocoronal direction;Class II: apico-coronal loss of tissue with normal ridge width in a bucco-lingual direction;Class III: combination of bucco-lingual and apico-coronal loss of tissue resulting in loss of height and width.

Immediate implant placement can be achieved without additional surgical treatment. However, slight hard tissue augmentation may be needed to add support to periimplant mucosa. There are situations that require soft tissue addition in order to aid maintenance [66,67]. There are several important factors that need to be taken into consideration during planning optimal placement of implants in the alveolar bone, that is, soft and hard tissue management, aesthetic factors, and proper quality of prosthetic restoration.

Conventionally, the placement of dental implants sacrifices much bone tissue during the drilling procedure. However, there are several implantation technique ideas that allow limited bone removal, especially in the case of patients with a limited amount of alveolar bone [68,69].

In order to perform a successful implantation, the dentist has to remember the periimplant values of hard and soft tissues. If the implants are placed too tightly, it will result in insufficient vertical blood supply to the papillae. Angulation of implants is crucial for a proper papilla development afterwards. There is no sufficient support for the papillae in two divergent crowns, while convergent crowns do not allow soft tissue to develop naturally. Tarnow et al. [70] demonstrated a proper relationship with regard to both implant to natural tooth and implant to implant. Regarding the former, in order to avoid horizontal bone loss, which will affect adjacent tooth, the distance should be about 2 mm. The latter requires a distance of at least 3 mm, which, when avoided, creates accelerated bone loss patterns in such areas [71]. It is important to underline that each implant loses periimplant bone within the first year and then stabilises [72].

The crown-to-implant ratio should be 1:1 or less, while the minimum height of the implant is 10–12 mm. The lower height of implant has already been proven to show a high failure rate [73]. In general, the actual height of bone is required to be 12 mm of bone actual height for a macroretentive screw-type implant to properly support occlusal forces [71].

### 4.3. Membranes in GBR

A GBR membrane acts as a barrier preventing fast-growing soft tissue from invading space required to be filled with a new bone [74]. Membrane materials for bone tissue engineering are usually divided into natural biomaterials [75], like chitosan, inorganic materials represented by nanohydroxyapatite [76], and synthetic polymer materials with polylactide-co-glycolide (PLGA) [77] as an example. Current approaches on graft materials exclusively face serious limitations [78]. Membranes themselves are not able to recover the defects in bone tissues completely, owing to their lack of satisfactory osteoinduction. The results change definitely with the incorporation of osteoinductive factors into semisolid or porous membranes-scaffolds [79,80].

Providing adequate space for bone regeneration is one of the fundamental principles of GBR. Various animal studies have proven that excluding the epithelium and connective tissue makes it possible to create that space, permitting slow migration of osteoblasts to the wound, which results in new bone formation [58,81]. Reinforced membranes allow the space maintenance by preventing membrane collapse that can occur owing to increased pressure from neighbouring tissues.

### 4.4. Resorbable Membranes

Degradable membranes can be made from collagen (natural), or poly (l-lactic acid) (PLLA) and PLGA [82] (polymeric). Many of these membranes are formulated with antibiotics, usually tetracyclines [83]. There are two important challenges in osteoinduction process that need to be overcome. The first one is to retain the osteogenic factor for a sufficient amount of time to generate the desired biological response, while the second concerns the biocompatibility of the material.

Achieving a desired tissue response is strictly dependent on both degradation components of the extracellular scaffold and concentration of inductive factors released from the matrix.

It is important to underline that bone can only grow when provided with space to do so. Thin, polymeric membranes allow to hold back soft tissues, but provide no significant mechanical support during bone healing. Polymeric adhesive or calcium phosphate cement are required for greater mechanical strength. In such a case, the bone is repaired alongside cement degradation, which can be a challenge.

One of the biggest advantages of resorbable membranes is the fact that they do not require a second surgical procedure over time. They undergo disintegration. On the other hand, it is important to underline, that such materials also have their limitations. Neiva R. et al. [84] have confirmed that using membranes to produce similar gains of keratinized tissue formation was failed in comparison with connective tissue grafts. Collagen material is resorbed by the host with the use of neutrophils and macrophages, causing no inflammation, while expanded polytetrafluoroethylene (ePTFE) material undergoes hydrolysis reaction, which, in some cases, may cause it.

### 4.5. Autogenous Bone Grafts

The “gold standard” bone graft material in traditional augmentation techniques is autologous bone graft. At the same time, allografts avoid donor site issues, but can cause a higher risk of infection and immune reaction of host tissue [85,86]. Autologous material is better than allograft, because it maintains bone structures, such as minerals, collagen, viable osteoblasts, and BMPs. Although autografts are a gold standard, they still present several significant limitations. Their harvesting is connected with the second concurrent surgical procedure, high donor site morbidity, and resorption [87,88].

Soft-tissue grafts are free gingival grafts that can be harvested from several sites in the patient oral cavity. In the reconstruction of minor alveolar defects, bone grafts from the retromoral region are one of the best intraoral sources possible [89,90,91]. Surgical operation concerning this region of oral cavity causes minimal discomfort for the patient, has relatively uncomplicated surgical access, and is in proximity of donor and recipient sites, which can lower the anesthesia doses required and cause only minor complications [92]. Its downside poses an unnecessary risk of complications owing to the involvement of the second surgical site. However, when the patient has inadequate thickness of palatal tissues, it is difficult to harvest a sufficient amount to place the graft properly.

Microvascular flap use is one of the states of the art and effective techniques for the repair of significant bone and soft tissue defects. While the vascularised pedicle allows proper perfusion to the harmed area, it also results in complete osseointegration of the bone graft [93]. On the other hand, even this technique represents several disadvantages—it requires a long intraoperative time for the patient, and may result in a permanent deficit, when a muscle or bone are included in the flap [94,95,96].

### 4.6. Allografts

Allograft is a tissue graft between individuals of the same specimen, but of nonidentical genetic composition. Generally, cadaver bone is a source for allografts, as it is available in large quantities [97]. One should realise and remember that the aforementioned bone has to undergo multiple treatment sequences in order to make it neutral for the host’s immune system and to avoid cross-contamination of disease. Allografts can be used as an alternative, but they have very limited osteogenicity and resorb more rapidly than autogenous bone. It is a useful material in case of patients requiring a non-union type grafting, who have inadequate autograft bone quantity, or when it is hard to obtain tissues from the donors’ site.

The main disadvantage of allografts is related to the relatively poor capacity for osteoconduction and osteoconduction, when compared with autologous graft. Another disadvantage of allografts is connected with an improper rate of resorption. It has to be clearly emphasised that bone tissue tends to be resorbed quickly, even after augmentation support, unless loading is provided with dental implant. Implants can be placed at the time of surgical procedure or 6 months later, after the stabilisation of the graft, which minimizes the resorption process.

One of the possible alternatives for soft-tissue grafts is acellular dermal matrix allografts (Alloderm). Alloderm is a donated human dermis, composed of a structurally integrated complex that constitutes basal membrane and an extracellular matrix [98]. Their use reduces the likelihood of cross-infection. Wagshall et al. [99] claim that if a graft material could be used to replace the palatal grafts, then all the possible complications connected to donor site would be immediately eliminated. This would result in alveolar ridge augmentation, being more acceptable for the dental patients.

### 4.7. Xenograft

Xenografts are a graft specimen from the inorganic portion of animal bones. Bovine is one of the most common sources for their extraction. In order to remove their antigenicity, the removal of the organic component is processed, whereas the remaining inorganic components both provide a natural matrix and serve as an excellent source of calcium. The disadvantage of xenografts is that they are only osteoconductive, and the resorption rate of bovine cortical bone is slow [15].

### 4.8. Bone Substitute Materials and Genetic Engineering

Genetic engineering can be done in two different ways. The first idea focuses on direct in vivo delivery of genes. There are reported cases in which recombinant human BMP-2 was mixed with an absorbable collagen sponge to treat open long-bone fractures [100]. Govender et al. [101] explained in their study that there was 44% reduction in the risk for failure in healing with less secondary invasive interventions and reduced healing time. The factor used during this study was recombinant BMP-2 called rhBMP-2/ACS. The second idea focuses on using autologous bone alternatives of either animal, human, or synthetic origin [102,103]. Such a technique has its limitations, for example, the risk of bacterial contamination [104], or effectiveness limited mainly to reconstruction of small bony defects [105].

## 5. Nanomaterials Application in Alveolar Bone Regeneration

### 5.1. Nanohydroxyapatite (n-HAp)

Nanomaterials, when compared with bulk materials, possess features like macroscopic quantum tunnelling or quantum size, causing altered physiochemical properties [106,107]. In oral biology, nanotechnology applications are mainly focused on augmentation procedures for osseous tissue regeneration and implants osseointegration enhancement [106]. Even though supraphysiological doses are necessary to combat the poor pharmacokinetics of this compounds, the nanocarriers can overcome such limitations by stabilizing the bioactive molecules.

Although bone autografts are considered the “gold standard” in clinical bone repair, they still have several limitations owing to the amount of bone that can be used, as well as an increased risk of donor site morbidity. Because of that, a considerable amount of study has been undertaken in order to develop effective regenerative strategies, leading to bone augmentation with limited side effects [108,109]. The n-HAp has a hierarchical architecture at multiple levels, including macrostructure (cancellous and cortical bone), microstructure (trabeculae), sub-microstructure (lamellae), nanostructure (embedded minerals and fibrillary collagen) [110], and sub-nanostructure (proteins and minerals). The presence of nanotubes or nanocrystals in the composite materials allows for enhancing the mechanical properties of the scaffolds. The nanosized materials present enhanced characteristics, like wettability, charge, roughness, and adsorption of proteins. Moreover, the nanotextured surfaces enhance in vitro osteogenesis and promote mineralization. Furthermore, in the case of the nanomaterials, aqueous contact angles become three times smaller, leading to increased adhesion of the osteoblasts in comparison with micro-sized materials.

Special composition and architecture allow them to have self-regenerative and self-remodelling ability in response to damaging signals and mechanical stimuli [111]. This makes nanomaterial an ideal candidate for bone graft development, as it is capable of recapitulating the organisation of the natural extracellular matrix, in order to regulate bone forming cells activity [112], as can be seen in Figure 4.

Synthetic biomaterials for bone repair should provide mechanical support and biological compatibility in order to promote bone tissue regeneration based on healing. Hydroxyapatite is an interesting inorganic mineral with potential dental [113], maxillofacial [114], and orthopaedic applications [115] that has a typical lattice structure as (A_10_(BO_4_)_6_C_2_) which defines A, B, and C by Ca, PO_4_^3−^, and OH^−^ [116]. Hydroxyapatite (Ca_10_(PO_4_)_6_(OH)_2_ is the principal inorganic mineral component of animal and human bones and teeth, and is difficult to dissolve in a solution where the ration of the calcium-to-phosphorus is 1:67 [117]. There are other forms of calcium phosphate present in nature, but HAp is the least soluble of them. The enamel is the hardest substance consisting of relatively large HAp and fluorapatite (FAp) crystals that are 25 nm thick, 40–120 nm wide, and 160 to 1000 nm long. In contrast to enamel, hydroxyapatite is present in bone as plates or needles, while its dimensions range from 40–60 nm long, 20 nm wide, and 1.5 to 5 nm thick.

Nano-HAp is a nanoform of hydroxyapatite with a range of unique properties and diameters ranging in size between 1 and 100 nm [118]. From these dimensions derives a distinct activity of the particles. It has been one of the most studied biomaterials in the medical fields, and has also proven to have strong biocompatibility [119], stability, and nontoxicity. Material possesses an ability of intense ion-exchange against various cations, causing HAp to have high bioactivity [120]. Diversity of n-HAp utility can be seen in Figure 5 and Table 1.

### 5.2. Examples of Bone Regeneration Using Nanohydroxyapatite and Stem Cells in Published Studies

Use of stem cells for bone healing and regeneration still remains in its infancy [121,122]. Composite grafts are able to incorporate osteogenic, osteoconductive and osteoinductive properties onto a compound [123,124]. For instance, local autogenous bone marrow can be harvested in order to combine with a bioceramic material. Also, use of antibiotic-loaded or antimicrobial bone graft substitutes has advantages over nonresorbable antibiotic carriers due to its biodegradability [125]. Nowadays, Nowadays, bone is the second most common transplanted tissue in comparison with blood tissue [126]. Implant bone graft-carrier allows to release the incorporated growth factor at the desirable rate and concentration. Additionally, it can be formed to be structured for facilitate cellular infiltration and growth [127]. 

Dahabreh et al. [128] check in their studies influence of bone graft substitutes on osteoprogenitor cells in terms of proliferation, differentiation and adherence. Moreover, Bojar et al. [129] confirmed effectiveness of alloplastic materials as an alternative for autologous transplants and xenografts in oral surgery and dental implantology.

The chemical composition most of them is hydroxyapatite. However, there is still a doubt to be successfully used in regenerative medicine. Although, the treatment using the allograft tissue is preceded by tissue freezing and freeze-drying as well as sterilization, there is always a risk of disease transmission from a donor [130] or rejection [131]. Furthermore, Kattimani et al. [132] proved that the limitations of allograft’s use with stem cells and nanohydroxyapatite has resulted in new alternatives. 

Nanosized bioceramics highly active surfaces and size make them a promising platform for bone regeneration. Such ceramics present increased osteoblast adhesion when compared with regular sized ceramics [133]. Their nanometre grain size is responsible for the increased osteoblast functions, like adhesion, proliferation, and differentiation. One of the examples of nanophase ceramics is n-HAp. It has been successfully used, among other applications, as a coating of orthopaedic implants, filler of composites, and bone filler [134]. According to several authors, the aforementioned material, in its pure form, is limited owing to its brittleness [135,136]. Wide ranges of solutions have been proposed in order to compensate problems with nanohydroxyapatite use, like incorporation of chitosan-biopolymer [137]. Wang et al. (2015) [138] proved, in their study, that scaffolds containing n-HAp, chitosan, and polylactide-co-glycolide (CS, PLGA) proved to have higher compression and tensile modulus, when compared with the same scaffolds that had no nanohydroxyapatite, which proves its important superior function. Bhyiyan et al. (2017) prepared a study in which they developed a multicomponent covalently-linked biodegradable biomaterial called n-HAp-PLGA collagen [139]. Its properties were similar to cancellous bone, and maintained high mechanical strength, even in an aqueous environment [139]. PLGA provides strong biodegradability, while hydroxyapatite bioceramic is responsible for osteoinduction/osteoconduction, while collagen allows biological stimulation for cell proliferation, similar to extracellular matrix in vivo [140]. PLGA is a synthetic biodegradable polymer. The main reaction used to create PLGA is ring opening polymerization and polycondensation of glycolic and lactic acids. On the basis of the studies of Tsai et al. (2010) [141], it can be confirmed that human mesenchymal stem cells’ (hMSCs’) growth on n-HAp-PLGA-collagen films is vast and comparable to collagen—a widely used substrate for hMSC attachment and proliferation [142]. Table 2 shows the versatile applications of n-HAp scaffolds in bone tissue regeneration.

### 5.3. Nanohydroxyapatite Doped with Rare Ions

In order to enhance n-HAp use in the regeneration field, it can be doped with specific materials. Evis et al. (2011) confirmed that the biodegradation rate of hydroxyapatite doped with metal ions was slower than that of pure mineral. The rate of resorption appeared to be minimal when the aforementioned material was doped with magnesium ions [143].

In implantology, absorption of the hydroxyapatite is crucial. It occurs simultaneously with replacement by bony tissue, and it can be achieved by matching implant resorption bone regeneration rates [144].

Qin et al. (2013) [145] demonstrated that osteogenic potential can increase as a result of the influence of Ag nanoparticles in human urine-derived stem cells, while also significantly increasing osteoblast lineage differentiation and mineralization in vivo [146]. On the other hand, Au particles also demonstrated promising application in both bone and cartilage repair [147].

Usually, one of the methods of n-HAp synthesis is the co-precipitation technique. In the case of Ag-HAp nanoparticles, it is similar, but with an addition of 1% Ag in the form of silver nitrate or 1% Au, respectively [148]. At the cellular level, n-HAp, Ag, and Au can be useful as possible promoters of osteogenic differentiation [149,150]. Hsu et al. (2007) [147] confirmed that Au nanoparticles enhance osteogenesis process through the mitogen-activated protein kinases signalling pathway, which can explain how Ag and Au can positively influence bone regeneration. n-HAp doped with such particles can be a representative of a powerful novel approach in the field of regenerative medicine.

### 5.4. Carbon Nanotubes as a Bone Regeneration Scaffold

Nowadays, a number of studies investigating the scaffold use of carbon nanotubes (CNTs) has been increasing [151,152]. In atomic scale, CNTs are hexagonal sheets of graphite wrapped into single or multiple sheets. They can be metallic or semiconducting, depending on their chirality. CNTs have thermal conductivity that is twice that of diamond and are stable up to 2800 °C. Its high bone affinity to serve as a scaffold makes it a promising material to use in regenerative medicine. CNTs possess outstanding mechanical properties, with their tensile strength in the range of 50–150 GPa, and a failure strain in excess of 5%. Several scientific works confirmed its effectiveness by checking osteoblasts adhesion to such complex [153], influence on proliferation of osteoblasts and osteocytes [151], and CNTs’ promotion of osseous tissue formation in vivo [154,155]. Tanaka et al. (2017) [151] confirmed that multi-walled CNTs (MWCNTs) blocks can serve as filler materials, because they are solid, with nano-sized surface irregularities and non-porous interiors, causing surrounding cells to not be capable of entering the scaffold. By allowing osteoblasts to proliferate on the MWCNT block surface, such a scaffold can have osteoconductive abilities. Their previous studies about CNT [156] additionally proved that this material can be successfully used as a functional scaffold for bone formation and promote the process of bone tissue regeneration.

### 5.5. 3D-Printed Scaffold Nanomaterials for Bone Application

Current surgical procedures for bone regeneration utilise transplantation using autografts, allografts, or xenografts, and have to deal with repair, renewal, and replacement of the bone tissue defect. Some of the main disadvantages of such operations are donor site morbidities, unavailability of large tissue volumes, additional risk of infections, communicable diseases, and severe pain [157,158].

Recently, 3D printing techniques appeared to be profitable as a tool for making scaffolds with controlled microarchitectures. They represent an attractive alternative for the synthesis of new scaffolds, allowing the modulation and control of the geometry with high precision over the pore size, outer shape of the scaffold, or porosity, all these together with cost-effectiveness and a rapid manufacturing process [159].

### 5.6. Graphene-Based Nanomaterial in Bone Regeneration

Graphene demonstrates the true sense of biomaterial by having two surfaces without bulk in between. Its single-atom-thick carbon-based honeycomb structure allows it to have uncharacteristically strong optical and mechanical electron properties. Currently, the methods such as the introduction of growth factors, genetic modifications, and cytokines have been used to help control stem cell differentiation [160]. Graphene is a 2D-nanostructure, which has similar mechanical, thermal, and electrical properties to carbon nanotubes and has potential for technological and scientific applications. The cytotoxic effects of graphene have been assessed, and it has been confirmed that layers made out of this material produce fewer toxic effects, which is crucial for bone regeneration. Low toxicity allows the final composite not to be rejected or induce an inflammatory reaction when placed inside the organism.

Biris et al. (2011) [161] have proven that nanocomposites can be synthesised in situ with a singular growth process, and they are characterised by high biocompatibility in osteoblastic bone cells’ proliferation in vitro. Graphene implementation in tissue engineering has offered unique scaffold structures with exceptional electrical and mechanical properties [162]. Its potential functionalization in combination with carbon backbone, nanoscale size, antibacterial activity [163] has been used as an enhanced method of controlled cell proliferation [164]. The recent biocompatible graphene nanocomposites can be prepared with the use of radio-frequency chemical vapour deposition (rf-CVD), using methane and acetylene as the carbon sources. Nanoclusters of such particles are evenly dispersed over HA with 2~7 nm diameters, and act as a catalyst for graphene synthesis [165]. The unique properties of nano-scale materials matched with cell sensitivity can be exploited to help improve the regeneration process [166].

Graphene-based HA nanocomposites can be prepared in the form of scaffolds, bulks, coatings, or powders. Both bulk composites and powders can be successfully used to repair the bone defects or small non-unions, as well as in coating metallic implants to increase bone-binding abilities. Porous graphene-based HA nanocomposites can be successfully used for larger bone defects. Such materials possess enhanced osteogenic activity and are promising when scaffolds are made out of nanomaterials. They also provide better outcomes in providing guided cell differentiation than in a situation when the cells are distributed directly into the defect. Nano-sized scaffolds are believed to better control the differentiation process, owing to their interaction with extracellular matrix (ECM) [162].

Loaded GFs, as well as adsorbed drugs on graphene and its derivatives, were able to increase osteogenic differentiation owing to increased local concentration. At the same time, the bone morphogenetic proteins (BMPs) are the most potent osteoinductive proteins for bone tissue regeneration [163].

### 5.7. Structural Effects of Nanomaterials on Bone Regeneration

At the time of bone regeneration, the porous architecture of scaffolds provides sufficient microenvironments for nutrient/waste exchange, cell proliferation, differentiation, and angiogenesis. The special structure of natural nanomaterials enhances bone with dynamic biological functions and mechanical durability. Nowadays, studies confirm that nanostructures at different dimensional levels play different roles in the regulation of bone regeneration [112]. For instance, during the initial period of implantation, biomaterials must provide structural support in the defect site for bone regeneration. Their nanofeatures act as a good enhancer to acquire proper mechanical properties and stability of osseous regeneration [165]. Nanoparticles can be incorporated into materials to form nanomaterials with adjustable mechanical strength, which can be used to induce stem cells’ osteogenic differentiation [166]. Lastly, nanoparticles alone can have the ability to improve osteogenesis for bone regeneration, for instance, Laponite [112]. In such materials, there is a possibility to adjust the conformation of GFs to increase their bioactivity for bone regeneration.

Nanoscaffolds have considerable drug loading abilities, high mobility of drug loaded particles, and efficient in vivo reactivity toward nearby tissues [167]. They can be used for labelling cells, in order to enable monitoring and continuous cell tracking [168], as well as enhancing osteoinduction, osteoconduction, and osseointegration [169].

Tissue engineering and regenerative medicine (TERM) aims to create functional substitutes for diseased and damaged tissues. The strategy behind TERM combines three essential elements, namely, scaffolds, GFs, and stem cells, as can be seen in Figure 6. Scaffolds provide support for tissue formation and are seeded with stem cells. GFs are also included as they regulate the differentiation and proliferation processes [169].

### 5.8. Hydrogels

Microengineering technologies can be successfully used in order to make hydrogel scaffolds mimic in vivo extracellular matrix (ECM). These techniques include lithography, micromolding, biopaterning, and microfluidics. According to Geckil et al. (2010) [170], hydrogels are three-dimensional, insoluble, cross-linked hydrophilic polymeric networks that are capable of providing scaffold to facilitate cell growth, infiltration, and differentiation [171], and can be used to deliver cells with regenerative functions, as pictured in Figure 7. Polymers in hydrogel can absorb a large amount of biological fluid or water with the help of interconnected microscopic pores. In order to increase the biological (hydrophilicity, cell-adhesiveness), mechanical (stiffness, viscoelasticity), and biophysical properties like porosity, combinations of either synthetic or natural hydrogels can be utilized. Such biomaterial composition makes them amenable to surface modification and biomimetic coatings. It is a type of polymer scaffold that has several potential advantages in bone repair. Hydrogels are materials that are able to mimic natural ECM of the bone, allowing to encapsulate bioactive molecules or cells. Network structure of the aforementioned materials allows proteins that are entrapped inside to be confined in the meshes of gel and released as required [172]. Moreover, those materials are absorbable and demonstrate magnificent integration with surrounding tissues, which removes the necessity of its surgical removal and additional trauma [173].

There are still some challenges that require further investigation. A controlled release of encapsulated drugs is one of them. Both burst and delayed release of the drug can affect actual therapeutic effect, and the use of inappropriate polymers can also cause toxic reactions [174]. Mimicking such a 3D-cell microenvironment in vitro with the use of hydrogels is crucial for various applications like constructing tissues for repair.

### 5.9. Nanostructured Scaffolds for Bone Tissue Engineering

Scaffolds can be utilised in bone tissue engineering in order to deliver biofactors including cells, genes, and proteins to generate bone and assessment of vascularity formation, together with overall tissue maturation [175]. There are three rules that a scaffold needs to comply with in order to be useful in tissue engineering. Firstly, it is required that the scaffold enhances the regenerative capability of the chosen biofactor; secondly, it must provide the correct anatomic geometry in order to maintain space for tissue regeneration; and thirdly, the scaffold needs to provide temporary mechanical load bearing within the specific tissue defect.

Many materials have been proposed as synthetic bone substitutes. Hydroxyapatite is regarded as one of the most bioactive bone substitute materials, mainly because of its superior osteoconductivity. On the other hand, synthetic octacalcium phosphate has been shown to be a good precursor of biological apatite in both teeth and bones, and it also presented better biodegradable and regenerative characteristics when compared with the other calcium phosphate bone substitute materials [176]. Hence, one of the disadvantages of such materials is their inability to achieve close apposition of the material to the neighbouring bony tissue, as well as brittleness of the ceramic materials. This can be overcome by mixing ceramic with, for example, polyesters, in order to form a composite that has good biodegradability, a high affinity for cells of polyesters, as well as osteoconductivity together with mechanical strength of calcium phosphates.

Mechanical properties can be enhanced by cross-linking. Arvidson et al. (2011) [175] give examples of polypropylene fumarate and CaSO_4_/-TCP materials that are similar to those of cancellous bone substitutes, with compressive strength of 5 MPa and modulus of 50 MPa during degradation [177].

Within the stem cell niche, nanoscale interactions with ECM components form another source of passive mechanical forces that can influence stem cell behaviours [178]. The ECM is built of a broad spectrum of structural polysaccharides and proteins that span over different length scales. Such a connection between stem cells and their nano-environment enables long-term maintenance and control of stem cell behaviour. The possibility to fabricate such small-scale technologies and platforms makes it possible to gain valuable insights into stem cell biomechanics [179].

Currently, scaffolds manufactured from nanotubes, nanoparticles, and nanofibres have emerged as promising candidates for better mimicking the nanostructure of natural ECM. They resemble it, and can be efficiently used to replace damaged tissues [110].

The ideal bone tissue scaffold should be osteogenic, osteoinductive, and osteoconductive [180]. The aforementioned biomimetic efforts include choosing biomaterials that are naturally present in bone, like collagen or hydroxyapatite. Other factors include incorporating growth factors like BMPs and fabricating multiple scale architectures in the scaffold.

It was confirmed by Gong et al. (2015) [178] and Kim et al. (2013) [181] that using nanogrooved matrices mimicking the natural tissues made it possible for the body and nucleus of hMSCs with the sparser nanogrooved pattern to become elongated and orientated more along the direction of nanogrooves than those with the relatively denser nanogroove patterns. The formation of cytoskeleton is crucial for the shape effect on the stem cell differentiation, while a type of synthetic ECM comprised of hierarchically multiscale structures can provide native ECM-like topographical cues for controlling the adhesion and differentiation of hMSCs. Interestingly, the platform that integrates hMSCs into the PLGA scaffold showed potential to regenerate the osseous tissues without the need for further surgical treatments.

Synthetically nanofabricated topography is also a factor that can influence the cell morphology, alignment, adhesion, migration, proliferation, and cytoskeleton organisation [182]. The conclusion is that there is an involvement of cytoskeleton into the stem cells’ physiology, suggesting the importance of the force balance along the mechanical axis of the ECM–integrin–cytoskeleton linkage, and their regulation by the mechanical signals in the stem cell niche [183]. Nowadays, there are additional possibilities of printing biocompatible scaffolds using the selective laser melting (SLM) method [184].

## 6. Tissue Engineering-Stem Cell Application in Bone Augmentation

Stem cells play vital roles in the repair of every tissue of the organism. They are undifferentiated cells, capable of renewing themselves and, by differentiation, they can be induced to develop into many different cell lineages [185]. They have been proved to be promising in tissue regeneration, as well as in the augmentation process. During bone reconstruction procedures, surgeons harvest autologous bone from the patient in order to transplant the graft to the injured site [186]. Moreover, autologous bone grafts still have an unpredictable resorption rate [187]. Nowadays, regeneration of large bony defects is still difficult to manage, despite many advances in bone regeneration treatment [37]. A tissue-engineering model is being promoted as an efficient, “state-of-the-art” technology for major osteogenesis [188,189].

In the view of increasing demands for bone grafting and limitations of “gold standard” procedures, surgeons are looking for a better approach. Tissue engineering allows combining synthetic scaffolds and molecular signals together with mesenchymal or bone marrow stem cells [175] to form hybrid constructs. The classical approach of tissue engineering concerns harvesting stem cells from the bone marrow; then isolating and expanding them; and, at the end, inserting the cells on a suitable synthetic or natural scaffold, before implantation into the same patient [190].

The goal of the modern approach is to reach stem cells present in more accessible sources in the human body, like periodontal ligament or deciduous and permanent teeth. According to Arvidson et al. (2011) [175], it is assumed that the perivascular region in the dental pulp is the niche for pulp-derived stem cells (PDSCs) of mesenchymal origin. In in vitro studies, PDSC are able to regenerate, and they have multilineage potential and plasticity [191] (including chondrocytes and osteoblasts).

Stem cell therapy demonstrates extraordinary value for many severe injuries and diseases. It includes key elements like extracellular matrix scaffolds and stem cells [192]. Furthermore, clinical trials about jaw bone regeneration applied in dental areas have demonstrated positive results [185].

Fortunately, different pre-osteogenic cells types can be used in the practice of bone regeneration. This type of cell is further differentiated into osteogenic cell lineages.

Nowadays, tissue engineering is a process focused on harvesting of multipotent stem cells from an autologous source and their successive in vitro culture. Then, the amount of such cells is increased within the injured tissue [193]. A major disadvantage of the stem cell transplantation is the need for large amounts of cells and accessibility.

There are different kinds of stem cells present in the human body; nevertheless, currently, there are only two main types of them that are used in clinical practice, namely mesenchymal stem cells (MSCs) and hematopoietic stem cells (HSCs).

### 6.1. Mesenchymal Stem Cells Use

MSCs are multipotent stromal cells that can differentiate into a variety of cells [194], including chondrocytes and osteoblasts. Even though bone marrow was the original source of MSCs, there are alternatives that have been drawn from the other adult’s tissues [195,196]. MSCs are nonhematopoietic, which results in their lack of contribution to the formation of blood cells like that of hematopoietic stem cells [197].

MSCs have been isolated from nearly every tissue possible, including brain, spleen, liver, kidney, lung, synovial membrane, or muscles [198,199,200,201]. They can be relatively easily expanded and differentiated into multiple tissue lineages, which makes them crucial in present and future tissue-engineering [202]. Another key feature of MSCs is their rapid expansion in vitro without loss of their characteristics. Bruder et al. [203] and Cancedda et al. [204] proved that bone marrow stem cells (BMSCs) are even capable of retaining their undifferentiated phenotype for 38 doublings, which eventually results in billion-fold expansion. MSCs transplanted systemically are able to migrate to specific site of injury in animals, which proves their migratory capacity.

They can be identified by the expression of several molecules, including CD105 (SH2) and CD73 (SH3/4). Additionally, they are negative for hematopoietic markers CD34, CD45, and CD14 [205].

Ashton et al. [206] have carried out an interesting experiment. They cultured freshly isolated rabbit marrow cells both in vitro and in diffusion chambers in vivo. The differentiation of osteogenic tissues in the diffusion chambers had to be divided into two categories: the first was the formation of bone in a fibrous layer surrounding cartilage, and the second was an intramembranous bone, formed directly within fibrous tissue that was not associated with cartilage. The authors suggested the presence of osteogenic precursors that had the potential to control the differentiation process via either of two major paths of skeletal development in embryo.

In the past, the MSC differentiation process in vitro involved incubating a confluent monolayer of MSCs together with β-glycerophosphate, ascorbic acid, and dexamethasone for 2–3 weeks [207]. The problem with the use of such factors in order to influence the cells was that it implausibly reflected physiological signals that MSCs received during osteogenesis in vivo. On the other hand, currently, the role of BMPs was investigated, which resulted in the promotion of bone growth in both humans and animals models [208].

### 6.2. Hematopoietic Stem Cells Use (BMMSCs)

Bone marrow-derived mesenchymal stem cells (BMMSCs) remain the most widely used osteogenic cells in bone tissue engineering research. They are present in adult bone marrow. They can be successfully used as an alternative for bone grafting, because of their immense replicative and differentiation capacity to form numerous connective tissue cells. BMMSCs can be isolated from the iliac crest [196]. Additionally, they can be obtained from orofacial bones, such as mandible and maxilla bone marrow suctioned during dental treatments (dental implantation), orthodontic osteotomy, cyst extirpation, or third molar extraction [209]. It has to be emphasised that, in both human and animal studies, the bone grafted from the craniofacial area was characterised with greater results and higher bone volume than when it was extracted from rib or iliac crest [210,211].

Mashimo et al. [212] positively evaluated alveolar ridge regeneration in the extraction sockets of mice. Stem cells were implanted immediately into the injured area in femur and tibia. Histological analysis proved that, after 3 and 6 weeks, the experimental group contained a greater quantity of bone marrow than the control group. BMMSCs have a number of advantages strictly connected to bone regeneration, including dynamic proliferation, relatively easy isolation, and expansion in vitro, as well as a lack of ethical controversy related to their medical use [213,214].

## 7. Conclusions

This review mainly focused on summing up the utility of nanomaterials and stem cells in maxillofacial bone tissue regeneration. Many recent scientific works in the fields of tissue engineering were investigated to demonstrate that both nanomaterials and stem cells can be successfully used as materials for guiding cell differentiation, proliferation, and organisation. It is important to underline that all the previously mentioned studies indicate the fact that nanomaterials alone may not be a complete answer to generate successful bone regenerative scaffolds. One of the primary challenges is to use innovative processing technologies in combination with nanomaterials. The construction of an ideal biomimetic nanocomposite would also require incorporation of a hierarchical design. Eventually, it could be possible to obtain an optimal scaffold in combination with several materials and techniques.

Recent advances in the fields of nanotechnology and tissue engineering have established great promise for finding treatments in bone defects and have led to considerable progress in designing and fabricating bone graft substitutes. For instance, impressive progress in the synthesis and functionalisation of graphene materials has opened up new possibilities for exploring their applications in tissue engineering. The primary function of an optimal biomaterial scaffold is to support the area undergoing reconstruction, providing adequate initial mechanical strength. It should also trigger a new bone formation, and later on gradually degrade, without causing an inflammatory response. The selection of the most appropriate scaffolding material is crucial in a tissue-engineered construct.

Scientific works focusing on stem cells have confirmed that mesenchymal stem cells derived from bone marrow are multipotential. They are attractive candidates for cell-based therapy, owing to self-renewal and immunosuppressive properties. Depending on culture conditions, they may differentiate into a plethora of cell types, including osteoblasts and chondrocytes. Thus, MSC comprise a readily available and abundant source of cells for tissue engineering applications. The lack of immunogenicity of MSC has opened up the potential of using those cells in tissue repair. The idea of such strategies is to take advantage of the body’s natural ability to repair injured bony tissue with new bone cells, and to remodel the newly formed tissue. Regardless of cell source, live cell-based implants appear to be superior to cell-free alternatives for bone tissue regeneration. Further research should be focused on developing techniques that combine both nanomaterials and stem cells therapies in order to allow even better clinical outcomes in the future.

## Figures and Tables

**Figure 1 nanomaterials-10-01216-f001:**
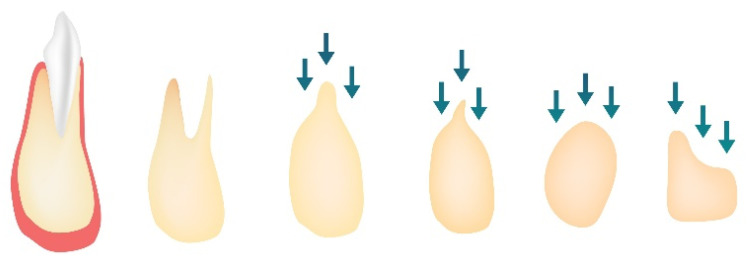
Stages of alveolar ridge reduction after tooth loss, order 1—pre extraction, order 2—post extraction, order 3—high, well-rounded, order 4—knife edge, order 5—low, well-rounded, order 6—depressed.

**Figure 2 nanomaterials-10-01216-f002:**
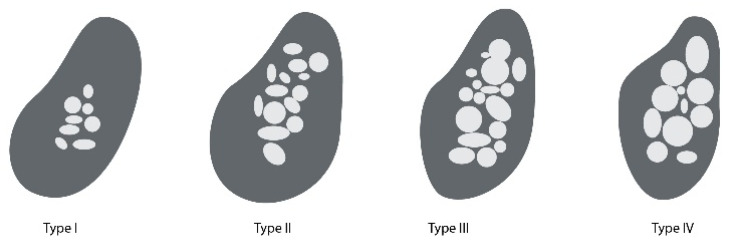
Lekholm and Zarb classification: Type I, whole bone is built of very thick cortical bone; Type II, thick layer of cortical bone surrounds the core of dense trabecular bone; Type III, thin layer of cortical bone surrounds the core of trabecular bone of good strength; Type IV, very thin layer of cortical bone with low density trabecular bone of poor strength.

**Figure 3 nanomaterials-10-01216-f003:**
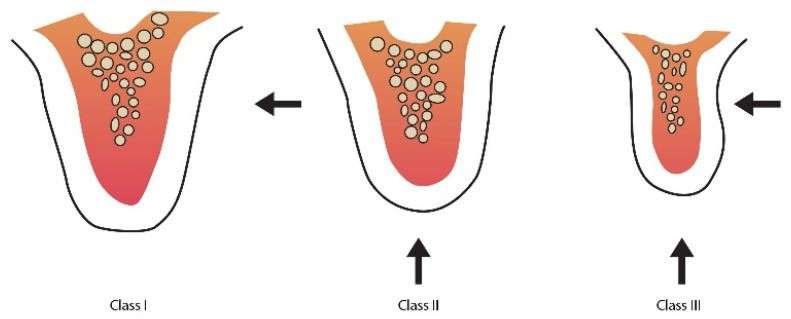
Ridge defect classification of edentulous patients according to Seibert (1983).

**Figure 4 nanomaterials-10-01216-f004:**
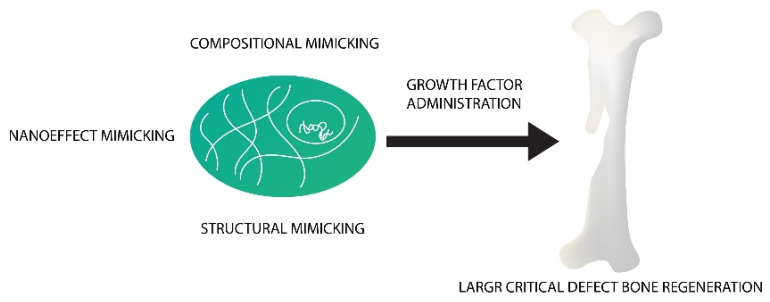
Schematic representation of how a synergistic combination of compositional, nanoelements, well-defined structure, and growth factor administration may endow nanomaterials with a “self-regenerative” capacity for the regeneration of large critical defect bone in a natural bone-healing way, especially at an in vivo level.

**Figure 5 nanomaterials-10-01216-f005:**
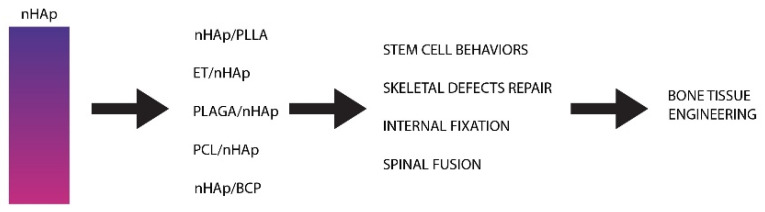
The application of nanohydroxyapatite (n-HAp) scaffolds in bone tissue engineering. PLGA, polylactide-co-glycolide.

**Figure 6 nanomaterials-10-01216-f006:**
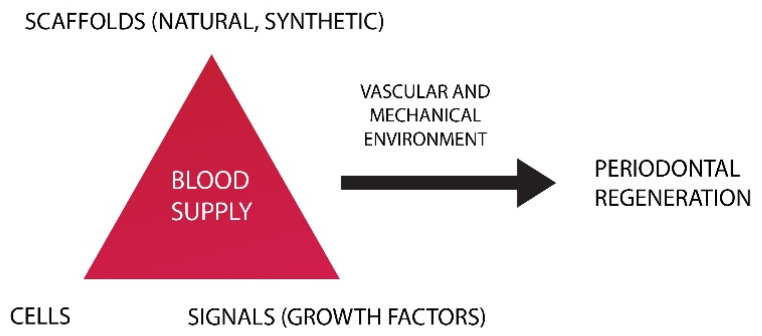
Tissue engineering triad.

**Figure 7 nanomaterials-10-01216-f007:**
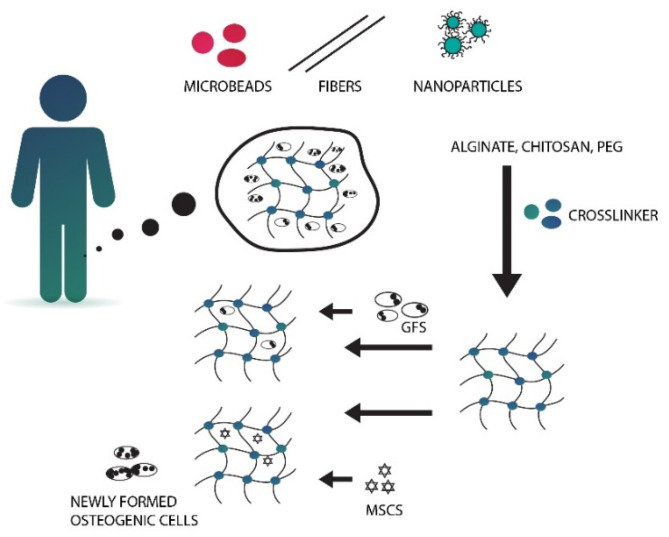
Schematic illustration of hydrogel-assisted bone regeneration. MSC, mesenchymal stem cell; GF, growth factor.

**Table 1 nanomaterials-10-01216-t001:** The application of n-HAp scaffolds in bone tissue engineering. BMP, bone morphogenetic protein.

Material	Application	Merit	Reference
Autograft	Spine fusion	Gold standard	[124,125]
Allograft	Craniofacial bone injury	Osteoinductive, osteoconductive	[126]
BMP	Open tibial fractures	Osteoinduction	[127]
Bioactive glass	Osteomyelitis	Anti-infective carrier	[128]
Composites	Femoral or cancellous bone defects	Biocompatible, tunable physiochemical properties	[129,130]
Synthetic polymers	Spine fusion, loading-bearing sites	Controlled degradation, mechanical strength	[131,132]
Natural polymers	Spinal fusion	Flexible, biocompatible, and biodegradable	[133]
Ceramic	Craniofacial bone defect	Biodegradable, osteointegrative, osteoconductive	[134]
Glass-ceramic	Femoral	Osteogenic	[135]

**Table 2 nanomaterials-10-01216-t002:** Types of membranes [87].

Resorbable Membranes	Non-Resorbable Membranes
Polylactic	Cellulose acetate filter
Polylactic/polyglycolic	PTFE
PL,PG and trimethylcarbonate	ePTFE
PG and TMC	Titanium mesh
Polyethylene glycol	Ethylene cellulose
Collagen	Rubber dam

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
