# Peer review of "Selected Nanomaterials’ Application Enhanced with the Use of Stem Cells in Acceleration of Alveolar Bone Regeneration during Augmentation Process"

_nanomaterials, 2020, doi:10.3390/nano10061216_

Round 1
Reviewer 1 Report
The topic of the review, acceleration of alveolar bone regeneration, is of high therapeutic relevance. However, the title of the review is not clear and does not match the content of the paper. The "chosen" is not clarified and I propose the authors to rephrase the title or delete the "chosen" word.
Several abbreviations occur in the text without resolution. The latter is disturbing and makes the text difficult to understand (e.g. Types of membranes).
Based on the title, the focus of the paper would be the role of nanomaterials in tissue engineering. Still, the article discusses the mechanism of tissue regeneration in detail, while the structure-activity relationships of nanomaterials are vague, in many cases, the molecular structure does not appear, only the abbreviation. The authors should add material characterization to the concerned subsections.
Author Response
Dear Editor,
We would like to express our sincerest gratitude to the Reviewers for their enormous efforts in criticizing the manuscript. We have taken into account all raised question here follows the detailed answers to the Reviewers. Moreover, all changes we have made to the original manuscript, are marked in the red colour in the text.
Reviewer 1
The topic of the review, acceleration of alveolar bone regeneration, is of high therapeutic relevance. However, the title of the review is not clear and does not match the content of the paper. The "chosen" is not clarified and I propose the authors to rephrase the title or delete the "chosen" word.
ANSWER: The topic has been changed.
Several abbreviations occur in the text without resolution. The latter is disturbing and makes the text difficult to understand (e.g. Types of membranes).
ANSWER: Correction of abbreviations has been made.
Based on the title, the focus of the paper would be the role of nanomaterials in tissue engineering. Still, the article discusses the mechanism of tissue regeneration in detail, while the structure-activity relationships of nanomaterials are vague, in many cases, the molecular structure does not appear, only the abbreviation. The authors should add material characterization to the concerned subsections.
ANSWER: Additional information of the reviewed materials has been added.

Reviewer 2 Report
In this manuscript the Authors gived an overview regarding clinical protocols and ongoing researches in bone regenerative medicine in the field of dental surgery and implants.
The manuscript appears well organized and rich of bibliographic sources.
Limits are found in the title (it is too long and not incisive, at least I suggest to modify it as “Selected nanomaterials' Application enhanced with the use of stem cells in the acceleration of alveolar bone regeneration during the augmentation process”) and in the Abstract (it appears without efficacy and without a clear message). The Abstract did not clearly underline the aim of the manuscript that did not represent a study but a critical review of research evidences and clinical results.
Author Response
Dear Editor,
We would like to express our sincerest gratitude to the Reviewers for their enormous efforts in criticizing the manuscript. We have taken into account all raised question here follows the detailed answers to the Reviewers. Moreover, all changes we have made to the original manuscript, are marked in the red colour in the text.
Reviewer 2
In this manuscript the Authors gived an overview regarding clinical protocols and ongoing researches in bone regenerative medicine in the field of dental surgery and implants.
The manuscript appears well organized and rich of bibliographic sources.
Limits are found in the title (it is too long and not incisive, at least I suggest to modify it as “Selected nanomaterials' Application enhanced with the use of stem cells in the acceleration of alveolar bone regeneration during the augmentation process”) and in the Abstract (it appears without efficacy and without a clear message). The Abstract did not clearly underline the aim of the manuscript that did not represent a study but a critical review of research evidences and clinical results.
ANSWER: The title has been changed.

Round 2
Reviewer 1 Report
The authors adequately addressed the reviewer' comments, therefore I suggest the publication of the paper.